# INFEKTA—An agent-based model for transmission of infectious diseases: The COVID-19 case in Bogotá, Colombia

Jonatan Gomez[1]*, Jeisson Prieto[2], Elizabeth Leon[1], Arles Rodríguez[3]

**1** Departamento de Ingeniería de Sistemas e Industrial, Facultad de Ingeniería, Universidad Nacional de Colombia, Bogotá, Colombia, **2** Departamento de Matemáticas, Facultad de Ciencias, Universidad Nacional de Colombia, Bogotá, Colombia, **3** Centro de Investigaciones de Matematicas e Ingenierías, Fundación Universitaria Konrad Lorenz, Bogotá, Colombia

* jgomezpe@unal.edu.co

## Abstract

The transmission dynamics of the coronavirus—COVID-19—have challenged humankind at almost every level. Currently, research groups around the globe are trying to figure out such transmission dynamics under special conditions such as separation policies enforced by governments. Mathematical and computational models, like the compartmental model or the agent-based model, are being used for this purpose. This paper proposes an agent-based model, called INFEKTA, for simulating the transmission of infectious diseases, not only the COVID-19, under social distancing policies. INFEKTA combines the transmission dynamic of a specific disease, (according to parameters found in the literature) with demographic information (population density, age, and genre of individuals) of geopolitical regions of the real town or city under study. Agents (virtual persons) can move, according to its mobility routines and the enforced social distancing policy, on a complex network of accessible places defined over an Euclidean space representing the town or city. The transmission dynamics of the COVID-19 under different social distancing policies in Bogotá city, the capital of Colombia, is simulated using INFEKTA with one million virtual persons. A sensitivity analysis of the impact of social distancing policies indicates that it is possible to establish a 'medium' (i.e., close 40% of the places) social distancing policy to achieve a significant reduction in the disease transmission.

## Introduction

Infectious diseases have a substantial impact on public health, health care, macroeconomics, and society. The availability of options to control and prevent the emergence, expansion, or resurgence of pathogens warrants continuous evaluation using different methods. Mathematical models allow characterizing both the behavior and the emergent properties of biological systems, such as the transmission of infectious disease. [1–3]. Many biological systems have been modeled in terms of complex systems since their collective behavior cannot be simply inferred from the understanding of their components [4, 5].

**Data Availability Statement:** Data used in this research is third party data (publicly available data) and can be obtained from the referenced public

sources. We did not have any special access or privileges to such data.

**Funding:** The author(s) received no specific funding for this work.

**Competing interests:** The authors have declared that no competing interests exist.

Computer-based algorithms are used to model properties and dynamic interactions between agents (e.g. persons, cells) or groups of agents within, and across levels of influence in complex systems [6, 7]. In general, agent-based modeling (ABM) can be used for testing theories about underlying interaction mechanics among the system's components and their resulting dynamics. It can be done by relaxing assumptions and/or altering the interaction mechanisms at the individual agent level. ABMs can increase our understanding of the mechanisms of complex dynamic systems, and the results of the simulations may be used for estimating future scenarios [8].

In the past, ABMs have been employed to address various infectious diseases such as, a bioterrorist introduction of smallpox [9], control of tuberculosis [10], implementation of distancing measures and antiviral prophylaxis to control H5N1 influenza A (bird flu) [11], design of vaccination strategies for influenza [12], devise evacuation strategies in the event of airborne contamination [13], and curtail transmission of measles through contact tracing and quarantine [14]. In our literature review, some other novel works that include heterogeneous agents and social distancing were proposed to model COVID-19 [15, 16]. The proposed approach, called INFEKTA (Esperanto word for infectious), mainly differs from existing works in that it aims to generate individuals and a complex network of places based on the population density of a determined city including individual interaction in public transportation means.

INFEKTA models the disease transition at the person level and takes into consideration individual infection disease incubation periods and evolution, medical preconditions, age, daily routines (movements from house to destination places and back, including transportation medium if required), and enforced of Non-Pharmaceutical Interventions such as social distancing policies may flatten the curve.

## Complex systems approaches for epidemic models

The complex system model approach considers a system as a large number of entities (equally complex systems that have autonomous strategies and behaviors) that interact with each other in local and non-trivial ways [17–19]. This approach provides a conceptual structure (a multi-level complex network [20]) that allows characterizing the interrelation and interaction between elements of a system and between the system and its environment [21]. In this way, a system is composed of sub-systems of second order, which in turn may be composed of sub-systems of the third-order [22]. Transmission dynamics of infectious diseases are not traditionally modeled at the individual level, but at the population-level with a compartmental model. However, some recent research use agent-based modeling for doing that [23].

## Compartmental model

A compartmental model tracks changes in compartments without specifying which individuals are involved [24] and typically reflects health states relevant for transmission (e.g., susceptible, exposed, infectious, and recovered). Basically, these kinds of models represent epidemics of communicable diseases using a population-based, non-spatial approach. The conceptual framework for this approach is rooted in the general population model which divides a population into different population compartments [25]. Compartmentalization typically reflects health states relevant for transmission (e.g., susceptible, exposed, infectious, and recovered, in short **SEIR**), though more partitioning is possible according to age and/or other relevant host characteristics. Heterogeneous and temporal behavior is modeled through the incorporation of relevant time-dependent social mixing, community structures, and seasonality, relevant for infectious disease dynamics [26, 27]. Process dynamics are captured in transition rates, representing the rate by which an average individual transitions between compartments.

## Agent-based models for infectious disease

Agent-based models (ABMs) are a type of computer simulation for the creation, disappearance, and movement of a finite collection of interacting individuals or agents with unique attributes regarding spatial location, physiological traits, and/or social behavior [23, 28, 29]. ABMs work bottom-up, with population-level behavior emerging from the interactions between autonomous individuals and their environment [23, 30]. They allow the history of every individual to be tracked and network structures to be explicitly represented.

In general, ABMs allow: i) To introduce local interaction rules at the individual level, which closely coincide with physical and social interaction rules; (ii) To include behaviors that may be randomized at the observational level, but can be deterministic from a mathematical point of view; (iii) To incorporate a modular structure and to add information through new types of individuals or by modifying current rules; and (iv) To observe systems dynamic that could not be inferred from the examination of the rules of particular individuals [8].

When ABM is used for epidemic modeling, infectious disease transmission dynamics is expected to emerge from the interaction between local interactions between the individuals. Each individual is modeled as an agent with an internal "SEIR" state that represents its infectious disease state (severity and time in it) at any instant of time. Individuals interact between them, i.e., can infect or get infected, when they move at some instant to the same place, their SEIR states, and the infectious disease transmission rates, and how close they are (if they are in crowded places). Notice, the concept of crowded places is natural in INFEKTA and emerges from the agents' interactions (eg. if more individuals move using the same transportation routes) and individual characteristics (e.g. children go to schools.). Transmission rates are usually approximated from the rates obtained by a compartmental model but are used at the individual level, i.e., when individuals interact with each other.

## INFEKTA agent-based model

Our agent-based model of infectious disease propagation, called INFEKTA, consists of five-layer components:

### Space

The virtual space (for a city or town being studied) is an Euclidean complex network [20]: Nodes are places (located in some position of the 2D Euclidean space) where individuals can be at some simulation time and edges are routes (straight lines) connecting two neighbor places.

- **Place (Node)**—A place may be of three kinds: home (where individuals live), public transportation station (PTS), and interest place (IP) i.e., school, workplace, market, and transportation terminal. IPs and PTSs are defined in terms of capacity (maximum number of individuals that can be at some simulation step time). IPs and PTSs may be restricted, during some period, to some or all individuals. Place restriction is established according to the social distancing rule that is enforced during such a period.

- **Neighbor (Edge)**—A PTS is a neighbor to another according to the public transportation system of the city or town being studied. Homes and IPs are considered neighbors to its closest PTS in the 2D Euclidean space. No home is neighbor to any other home neither an IP is neighbor of any other IP. Finally, a home and an IP are considered neighbors if they are neighbors of the same PTS. Each individual has a Home and an IP. The closest distances are computed between each Home and IP and between these places with the closest PTS using their longitude and latitude. If the distance between a Home and IP is shorter than the

distance to a PTS this Home will be connected directly to an IP instead of their closest PTS. Otherwise, the closest PTS is connected to each Home and IP respectively. Detailed information can be found in Fig 2 of Section Virtual Space Setup.

## Time

Virtual time is defined in INFEKTA at two resolution levels: days for modeling the transmission dynamics of the infectious disease, and hours for modeling the moving and interaction of individuals. Therefore, if an individual gets infected more than once during the same day, INFEKTA considers all of them as a single infection event. Any individual movement is carried on the same one hour, it was started, regardless of the traveled Euclidean distance neither the length of the path (number of edges in the complex network).

## Individuals

A virtual individual in INFEKTA is defined in terms of his/her demographic, mobility, and infectious disease state information.

- **Demographics**—The demographic information of a virtual individual consists of: **i) Age** of the individual; **ii) Gender** of the individual *female* o *male*; **iii) Location** of the individual at the current time step; **iv) Home** of the individual, **v) Impact level of medical preconditions** on the infectious disease state if the individual is infected, and **vi) IP interest** of going to certain type of IPs.

- **Mobility**—The ability of an individual to move through space (we use the graph defining the space for determining the route as proposed in [31, 32]). Each individual has a **mobility plan for every day**, plan that is carried on according to the enforced social distancing policy and her/his infectious disease state. The mobility plan is modeled in INFEKTA as a collection of simple movement plans to have **i) Policy**: social distancing policy required for carrying on the mobility plan; **ii) Type**: may be *mandatory*, i.e., must go to the defined interest place) or *optional*, i.e., any place according to individual's preferences; **iii) Day**: day of the week the plan is carried on, maybe *every, week, weekend, Monday, . . ., Sunday*; **iv) Going Hour** time an individual moves from Home to an IP; **v) Duration** in hours for coming back to home, and **vi) Place**: if plan type is mandatory, it is a specific place, otherwise it is an IP selected by the individual according to his/her IP preferences.

## Infectious diseases dynamic

Fig 1 shows the general transition dynamics of any infectious disease at the individual level in INFEKTA. This model can be adjusted to any specific infectious disease by setting some of the probabilities to specific values. For example, if there is no evidence that recovered individuals become immune or susceptible again, such probabilities can be set to 0.0.

Any individual can potentially be in one of seven different infectious disease states or health states in INFEKTA: Immune ($M$), Susceptible ($S$), Exposed ($E$), Asymptomatic-Infected ($I_A$), Seriously-Infected ($I_S$), Critically-Infected ($I_C$), Recovered ($R$), Dead ($D$), and Immune ($M$). As can be noticed, we just adapt the terminology from the compartmental models in epidemiology—namely, from the SEIR (Susceptible-Exposed-Infectious-Recovered) model. In INFEKTA, the infectious state of the SEIR model is divided into asymptomatic-infected, seriously-infected, and critically-infected in order to capture how age, gender, IP preferences,

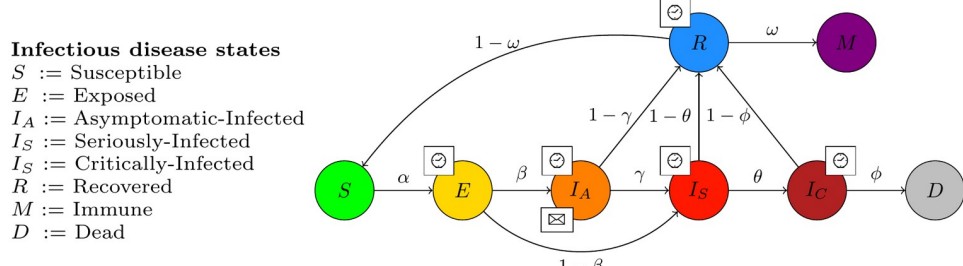

**Infectious disease states**
$S$ := Susceptible
$E$ := Exposed
$I_A$ := Asymptomatic-Infected
$I_S$ := Seriously-Infected
$I_S$ := Critically-Infected
$R$ := Recovered
$M$ := Immune
$D$ := Dead

**Fig 1. General transmission dynamics of any infectious disease at individual level in INFEKTA.** Probabilities are individual based and are defined according to the infectious diseases and characteristic such as current location, age, gender, and so on. Symbol ⟳, on state $X$, indicates that an individual must stay some period of time $T_X$ at such state $X$ before being able to change to other state. Symbol ✉, indicates that individuals on state $X$ can infect Susceptible ($S$) individuals.

medical preconditions (co-morbidity), and social distancing policies can impact the evolution of the infectious disease in an individual. INFEKTA introduces both the $M$ state since some individuals are naturally immune to or can become immune to (after recovering) to certain infectious diseases and the Dead ($D$) state to distinguish between recovered and dead individuals.

Since rates are defined at the individual level, these rates can be defined by taking into consideration, for example, rates at the population level (obtained from a compartmental model), age, gender, and co-morbidity presented in the individual. Remember that those rates are not defined at some time scale (as in compartmental models) but define the rule determining changes in health states of individuals being close enough for interacting at the infectious disease transmission level or after some period of time being in some state. In this way, an individual can change with probability $\alpha$ from $S$ state to state $E$ if close enough to an asymptomatic-Infected ($I_A$) individual and will change from state $I_A$ to state $I_S$ if has been at state $I_A$ with probability $\theta$ if has been on state $I_A$ a period of time $T_{I_A}$.

- $\alpha$: is the transmission rate and incorporates the encounter rate between susceptible and infectious individuals together with the probability of transmission.

- $\beta$: is the rate at which individuals move from the exposed ($E$) to the Asymptomatic-Infected state ($I_S$). It's complement ($1 - \beta$) is the rate of individuals with symptomatic cases.

- $\gamma$: is the rate at which individuals move from the exposed ($I_A$) to the Seriously-Infected state ($I_S$).

- $\theta$: is the rate at which individuals move from the Seriously-Infected ($I_S$) to the Critically-Infected state ($I_C$).

- $\phi$: is the death rate.

- $\omega$: is the immune rate that incorporates the probability of becoming immune.

- $T_E$: Time an individual will be at the Exposed ($E$) state before changing to the Asymptomatic-Infected ($I_A$) or Seriously-Infected ($I_S$) states.

- $T_{I_A}$: Time an individual will be at the Asymptomatic-Infected ($I_A$) state before changing to the Seriously-Infected ($I_C$) or Recovered ($R$) states.

- $T_{I_S}$: Time an individual will be at the Seriously-Infected state ($I_S$) before changing to the Critically-Infected ($I_C$) or Recovered ($R$) states.

- $T_{I_C}$: Time an individual will be at the Critically-Infected ($I_C$) before changing to the Dead ($D$) or Recovered ($R$) states.

- $T_R$: Time an individual will be at the Recovered state ($R$) before changing to the Immune ($M$) or Susceptible ($S$) states.

INFEKTA can consider that two individuals were close enough for interacting at the transmission of the infectious disease if they were at the same place (home, interest place, or public transportation station) at the same time. In order to simplify this checking process, it is possible to consider that an individual just visited its home, final interest place, and both the initial and final PTSs when using the public transportation system.

### Social distancing policy

The social distancing policy is described in INFEKTA as a finite sequence of rules, each rule having **i) Start Time**: an initial day for applying the social distancing policy rule; **ii) End Time**: final day for ending the social distancing policy rule; **iii) Level**: indicates the kind of restriction applied to the mobility of persons and accesses to places, and **iv) Enforce**: defines the specific mobility and access restrictions of the social distancing policy.

## Modeling transmission dynamics of the COVID-19 in Bogotá—Colombia

INFEKTA is used for modeling the Transmission dynamic of the COVID-19 in Bogotá city, the largest and crowded city in Colombia. Bogotá is the capital city of Colombia, its urban perimeter population is 7.412.566, is composed by 112 Zonal Planning Units (*UPZ*, for its acronym in spanish). Each *UPZ* belongs to one of the 19 urban *districts* in Bogotá. Also, Bogotá massive public transportation system is called *Transmilenio* (**TM**). TM is a bus-based system, which has 143 stations and moves near to 2.500.000 citizens every day. Data used in this research is third party data (public available data) and can be obtained from the referenced public sources [33, 34]. We did not have any special access or privileges to such data.

### Virtual space setup

Geographical information of Bogotá is used as the Euclidean space where the moving and interaction complex network is defined. Each one of the TM stations is located and added to the complex network according to the real TM system [33]. Also, the airport and the regional bus terminal are located and connected to the nearest TM station.

Demographic information from 112 *UPZ* is used for generating in the Euclidean space interest places (Workplaces (W), markets (M), and schools (S)), homes (H), and people(P). Places are generated, in each one of the districts, following a 2D multivariate normal distribution $N \sim (\mu, \Sigma)$ ($\mu$ is the geographic center of the *UPZ* and $\Sigma$ is the co-variance matrix defined by the points determining the perimeter of the district). The number of places in each *UPZ* is generated based on the population density of each UPZ according to the data available in 2017 [34]. Table 1 shows the amount of data generated for each type of place and for people, also the number of TM stations (Bus), and terminal transportation that we use in the simulation, and Table 2 shows detailed information of the number of interest places generated by UPZ.

**Table 1. Data used in the simulation.**

| Agent | Instance | Type | Amount |
|---|---|---|---|
| Place | Home (H) | Home (H) | 297260 |
| | Public Transportation Station (PTS) | Bus (B)* | 143 |
| | Interest place (IP) | Workplace (W) | 118952 |
| | | School (S) | 59483 |
| | | Market (M) | 98126 |
| | | Terminal (T)* | 2 |
| Individual | Individual | People (P) | 998213 |

(*) real places.

Fig 2 shows an example of 1000 virtual places in the Euclidean map of Bogotá [35]; also, the figure shows the associated complex network of connected places (nodes are places and edges are routed between places), the graph was drawn with Gephi [36].

## Individuals setup

An heterogeneous (varying gender, age, district and home) group of almost one million of individuals (998213) is generated using a stratified sampling based on the demographic information of the city for each district according to the projections to 2030 [34]. An individual is classified, according to her/his age, as: *Child* = [0-9], *Adolescence* = [10-19], *Adult* = [19-49],

**Table 2. Number of interest places generated by *UPZ* group by *District*.**

| DISTRICT | Amount of places | | | |
|---|---|---|---|---|
| | Homes(H) | Markets(M) | Schools(S) | Workplaces(W) |
| (01) Usaquén | 19135 | 6307 | 3830 | 7625 |
| (02) Chapinero | 5410 | 1801 | 1100 | 2184 |
| (03) Santa Fe | 3944 | 1284 | 790 | 1578 |
| (04) San Cristóbal | 16806 | 5526 | 3367 | 6725 |
| (05) Usme | 14600 | 4840 | 2930 | 5875 |
| (06) Tunjuelito | 9106 | 2966 | 1817 | 3597 |
| (07) Bosa | 23103 | 7656 | 4650 | 9253 |
| (08) Kennedy | 40797 | 13452 | 8135 | 16322 |
| (09) Fontibón | 13458 | 4447 | 2688 | 5374 |
| (10) Engativá | 34428 | 11402 | 6906 | 13787 |
| (11) Suba | 41045 | 13517 | 8200 | 16423 |
| (12) Barrios Unidos | 9381 | 3101 | 1864 | 3776 |
| (13) Teusaquillo | 5896 | 1912 | 1163 | 2336 |
| (14) Los Mártires | 3921 | 1300 | 798 | 1581 |
| (15) Antonio Nariño | 4485 | 1503 | 885 | 1819 |
| (16) Puente Aranda | 10462 | 3454 | 2097 | 4189 |
| (17) La Candelaria | 1174 | 395 | 239 | 460 |
| (18) Rafael Uribe | 15282 | 5048 | 3058 | 6108 |
| (19) Ciudad Bolívar | 24827 | 8215 | 4966 | 9940 |
| **TOTAL** | **297260** | **98126** | **59483** | **118952** |

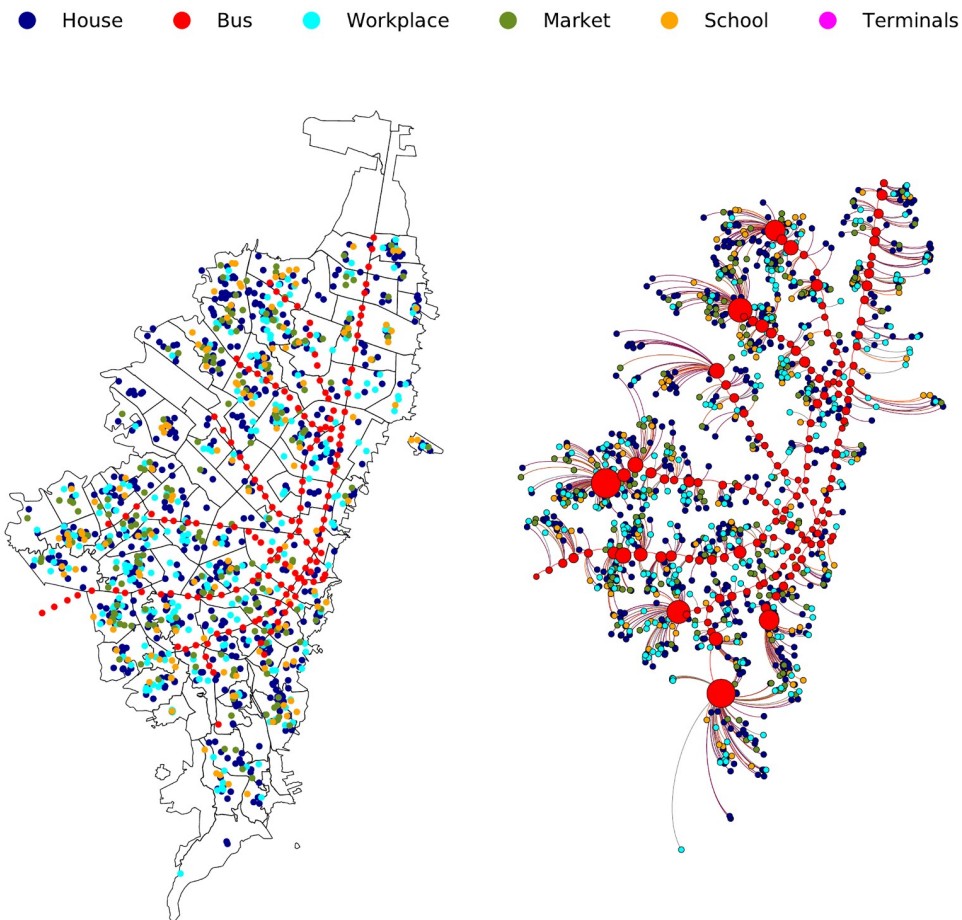

**Fig 2. Example of 1000 georeferenced places in Bogotá (left) and it's corresponding representation in the euclidean complex network (right).**

*Senior* = [50-69], and *older* = 70+. Table 3 shows the total demographic information of virtual people.

Also, a sequence of activities was assigned randomly to each individual to define a diary routine. This was done according to the person's age and the hour of the day. For example, some agents *Adolescence* go to school, and some agents *Adult* go to work. Time to start routine -going from Home to IP and return- is randomly selected in the interval from 4h and 7h returning between the 17h and 20h. Some agents may move using the PTI system and some others while going directly to its destination place. The route an individual takes is defined according to the complex network. Fig 3 shows three examples of different routines (paths over the graph) for the individuals.

The explicit impact level of medical preconditions on the state of the COVID-19 dynamic is not included in this preliminary modeling. We wrapped them in the transition rates and allow modelers to change and play with different rates. Therefore, we set the initial values of these rates as shown in Table 4.

## Social distancing rule setting

The level attribute of the social distancing rule for the COVID-19 in the virtual Bogotá city is defined as follows:

**Table 3. Demographic information of virtual people grouped by *District*.** District (D), Total(T), Male(M), Female(F).

| D | TOTAL | | | Child [0-9] | | | Adolescence [10-19] | | | Adult [19-49] | | | Senior [50-69] | | | older 70+ | | |
|---|---|---|---|---|---|---|---|---|---|---|---|---|---|---|---|---|---|---|
| | T | M | F | T | M | F | T | M | F | T | M | F | T | M | F | T | M | F |
| (01) | 62358 | 33485 | 28873 | 11143 | 5601 | 5542 | 4599 | 2337 | 2262 | 28561 | 15163 | 13398 | 14157 | 7990 | 6167 | 3898 | 2394 | 1504 |
| (02) | 17459 | 9272 | 8187 | 2176 | 1075 | 1101 | 1199 | 598 | 601 | 8473 | 4410 | 4063 | 4148 | 2301 | 1847 | 1463 | 888 | 575 |
| (03) | 13145 | 6541 | 6604 | 2930 | 1371 | 1559 | 1076 | 509 | 567 | 5774 | 2856 | 2918 | 2554 | 1340 | 1214 | 811 | 465 | 346 |
| (04) | 51501 | 26341 | 25160 | 13343 | 6472 | 6871 | 4628 | 2269 | 2359 | 22840 | 11683 | 11157 | 8499 | 4625 | 3874 | 2191 | 1292 | 899 |
| (05) | 55236 | 27907 | 27329 | 16168 | 7801 | 8367 | 5378 | 2621 | 2757 | 24251 | 12321 | 11930 | 7940 | 4286 | 3654 | 1499 | 878 | 621 |
| (06) | 25394 | 12793 | 12601 | 5865 | 2791 | 3074 | 2228 | 1072 | 1156 | 11582 | 5815 | 5767 | 4340 | 2316 | 2024 | 1379 | 799 | 580 |
| (07) | 82103 | 41953 | 40150 | 22635 | 11015 | 11620 | 7200 | 3542 | 3658 | 37699 | 19349 | 18350 | 12313 | 6712 | 5601 | 2256 | 1335 | 921 |
| (08) | 135750 | 69517 | 66233 | 32179 | 15606 | 16573 | 10909 | 5349 | 5560 | 63480 | 32467 | 31013 | 24067 | 13080 | 10987 | 5115 | 3015 | 2100 |
| (09) | 48288 | 25416 | 22872 | 10311 | 5122 | 5189 | 3665 | 1840 | 1825 | 23352 | 12250 | 11102 | 8843 | 4923 | 3920 | 2117 | 1281 | 836 |
| (10) | 111026 | 57864 | 53162 | 22632 | 11102 | 11530 | 8331 | 4132 | 4199 | 52205 | 27028 | 25177 | 21980 | 12094 | 9886 | 5878 | 3508 | 2370 |
| (11) | 149078 | 78324 | 70754 | 33063 | 16409 | 16654 | 11747 | 5894 | 5853 | 70984 | 37201 | 33783 | 26900 | 14964 | 11936 | 6384 | 3856 | 2528 |
| (12) | 30578 | 15877 | 14701 | 5002 | 2418 | 2584 | 2052 | 1003 | 1049 | 13456 | 6867 | 6589 | 7579 | 4113 | 3466 | 2489 | 1476 | 1013 |
| (13) | 19180 | 10230 | 8950 | 2428 | 1205 | 1223 | 1283 | 642 | 641 | 8958 | 4680 | 4278 | 4812 | 2674 | 2138 | 1699 | 1029 | 670 |
| (14) | 12536 | 6210 | 6326 | 2365 | 1099 | 1266 | 947 | 445 | 502 | 5670 | 2776 | 2894 | 2742 | 1428 | 1314 | 812 | 462 | 350 |
| (15) | 13827 | 7090 | 6737 | 2999 | 1445 | 1554 | 1117 | 544 | 573 | 5968 | 3032 | 2936 | 2826 | 1528 | 1298 | 917 | 541 | 376 |
| (16) | 32801 | 16654 | 16147 | 6186 | 2948 | 3238 | 2474 | 1192 | 1282 | 15382 | 7731 | 7651 | 6563 | 3509 | 3054 | 2196 | 1274 | 922 |
| (17) | 3059 | 1434 | 1625 | 483 | 212 | 271 | 278 | 123 | 155 | 1369 | 633 | 736 | 731 | 360 | 371 | 198 | 106 | 92 |
| (18) | 47610 | 24149 | 23461 | 11532 | 5536 | 5996 | 4113 | 1995 | 2118 | 21440 | 10850 | 10590 | 8312 | 4471 | 3841 | 2213 | 1297 | 916 |
| (19) | 87284 | 44552 | 42732 | 25410 | 12375 | 13035 | 8172 | 4024 | 4148 | 38710 | 19868 | 18842 | 12614 | 6883 | 5731 | 2378 | 1402 | 976 |
| | 998213 | 515609 | 482604 | 228850 | 111603 | 117247 | 81396 | 40131 | 41265 | 460154 | 236980 | 223174 | 181920 | 99597 | 82323 | 45893 | 27298 | 18595 |

- None: No restrictions to the mobility neither to access to places.

- Medium: Many places and few stations are restricted (depending on the type, capacity, etc). Some type of individuals is restricted to stay at home (except those with the required mobility level). i.e., close 40% of the places.

- Extreme: Few places are accessible to persons while few stations are restricted. Almost every individual is restricted to stay at home (except those with the required mobility level). i.e., close 80% of the places.

## Results

The methodology (Data preprocessing; places, population, and routes assignation; network creation) is available in a Github repository, see S1 File. We run a total of 20 experiments and the results (COVID-19 dynamics, sensitive analysis, and social distance policies) show below are the mean of those experiments.

Fig 4 shows the state of the COVID-19 dynamics (propagation of the virus) after 100 days when at the beginning of the simulation, just fifty (50) individuals are considered at state Asymptomatic-Infected ($I_A$). Fig 4 (left) shows the full dynamics over time. Notice that, using the parameters reported in the literature, the dynamics of the COVID-19 pandemic shows one wave. As expected, the number of asymptomatic, seriously, and critically infected cases grows exponentially when a social distancing policy is not enforced. Fig 4 (right) shows the COVID-19 dynamics at specific ticks (i.e., after 34 days—February 03 and after 67 days—March 07). For each tick, a map of Bogotá with the dynamics is presented. The map shows the percentage

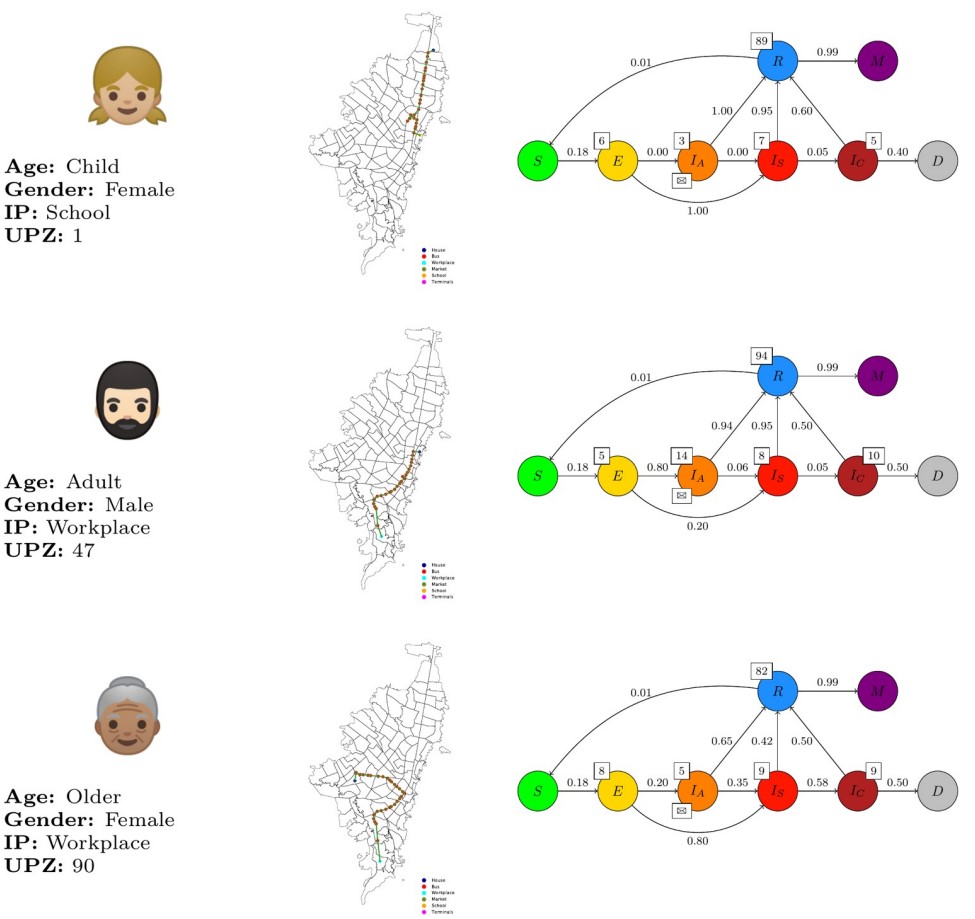

**Fig 3. Example routines carried by individuals.** Individual 38 (top): [Child, F, School, 1]; Individual 73128 (middle): [Adult, M, Workplace, 47]; Individual 349915 (bottom): [Older, F, Workplace, 90].

**Table 4. Parameters of INFEKTA and their estimations for COVID-19.**

| Symbol | Description | COVID-19 estimations | | | | | References |
|---|---|---|---|---|---|---|---|
| | | Child | Teen | Adult | Senior | Older | |
| $\alpha$ | Probability of $S \rightarrow E$ | 0.180 | | | | | [15, 37] |
| $\beta$ | Probability of $E \rightarrow I_A$ | 0.000 | 0.800 | | 0.200 | | [38] |
| $\gamma$ | Probability of $I_A \rightarrow I_S$ | 0.000 | 0.008 | 0.058 | 0.195 | 0.350 | [39] |
| $\theta$ | Probability of $I_S \rightarrow I_C$ | 0.050 | 0.050 | 0.050 | 0.198 | 0.575 | [15] |
| $\phi$ | Probability of $I_C \rightarrow D$ | 0.400 | | 0.500 | | | [15] |
| $\omega$ | Probability of $R \rightarrow M$ | 0.999 | | | | | - |
| $T_E$ | Time (days) at $E$ | $\Gamma(\alpha = 5.100, \beta = 0.860)$ | | | | | [40] |
| $T_{I_A}$ | Time (days) at $I_A$ | 3 | 14 | | 5 | | [41] |
| $T_{I_S}$ | Time (days) at $I_S$ | Triangular(7, 8, 9) | | | | | [42, 43] |
| $T_{I_C}$ | Time (days) at $I_C$ | Triangular(5, 7, 12) | | | | | [42] |
| $T_R$ | Time (days) at $R$ | $U(80, 100)$ | | | | | - |

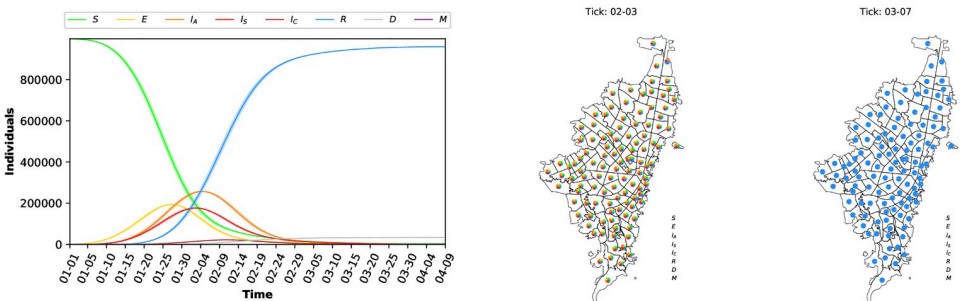

**Fig 4. Evolution of the epidemic dynamics of INFEKTA.** Full dynamics over time (left), and dynamics for each *UPZ* at specific time (right).

of individuals at each state per *UPZ*. After 34 days, the majority of individuals have been Exposed, a bunch of individuals are in the infected states (asymptomatic, seriously, and critically) and few of them become recovered; After 67 days, all individuals have been exposed to the virus and stay in the recovered state. Clearly, the transmission dynamics of the infectious disease (in this case COVID-19) emerge from the local interaction of the individuals.

We analyze the sensitivity to the infection rate ($\alpha$) parameter to check the robustness of the model. Sensitivity analysis is shown in Fig 5. Notice that by increasing or decreasing the infection disease rate (Fig 5 (left)), the peak of the transmission dynamics is reached sooner or later on time. When low infection disease rates, the number of cases is also low, reducing the impact on the economy. On the other hand, for high infection disease rates (Fig 5 (right)) the peak is reached in an early stage, and around half of the population is on one of the infected states (Asymptomatic, Seriously, Critically).

Also, we try different scenarios where each one of the social distancing policies is enforced just after 15 simulation days, see Fig 6. As can be noticed, it's evident how social distancing rules help to mitigate the exponential growth on the transmission disease dynamics (COVID-19), reducing the number of infectious cases (Asymptomatic, Seriously, and Critically).

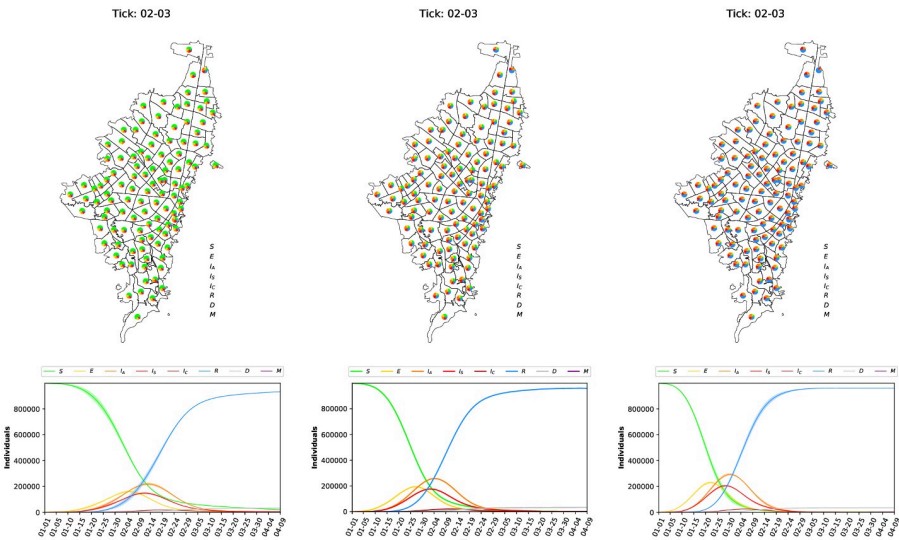

**Fig 5. Sensitive analysis for the infection probability $\alpha$.** $\alpha = 0.09$ (left), $\alpha = 0.18$ (middle), $\alpha = 0.36$ (right).

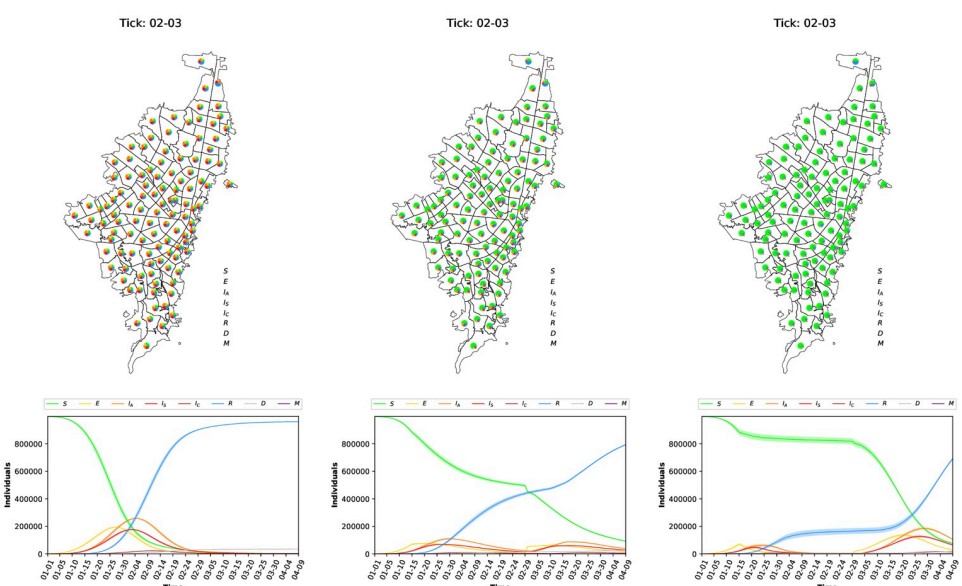

**Fig 6. Social distancing after 100 days when initialized at day 15 and ended at day 60 of the simulation.** None—No restrictions (left), Medium—close 40% of places (middle), and Extreme—close 80% of places (right).

Interestingly, when the extreme social distancing rule (access to approximately 80% of interest places is restricted) the transmission disease dynamic displays a big second wave with more cases than the first wave.

Although our intention was not to predict geographic spread for the city, we observed similarities between the total Seriously-Infected cases in INFEKTA (we assume that individuals in Seriously-Infected state are cases tested in Bogotá) and the current concentration of COVID-19 cases confirmed in Bogotá [44], see Fig 7. The results show how UPZ with more cases found with INFEKTA matches with geographic areas with more COVID-19 cases. Then, how population distribution is generated from real density data of Bogotá, INFEKTA can be useful to explore policies by Zonal Planning Units (UPZ) or territorial divisions of a selected place providing to recommend actions for before, during, and after pandemic i.e., in planning and coordination efforts through leadership and coordination across sectors.

## Conclusions and future work

Modeling the Transmission dynamic of an infectious disease such as the COVID-19 is not an easy task due to its highly complex nature. When using an agent-based model, several different characteristics can be modeled, for example, the demographic information of the population being studied, the set of places and the mobility of agents in the city or town under consideration, social distancing rules that may be enforced, and the special characteristics of the infectious disease being modeled. INFEKTA is an agent-based model that allows researchers to combine and study all of those characteristics.

Our preliminary results modeling the transmission dynamics of the coronavirus COVID-19 in Bogotá city, the largest and crowded city in Colombia, indicate that INFEKTA may be a valuable asset for researchers and public health decision-makers for exploring future scenarios when applying different social distancing policy rules and controlling the expansion of an infectious disease. Although we are doing a rough and no so real approximation of the

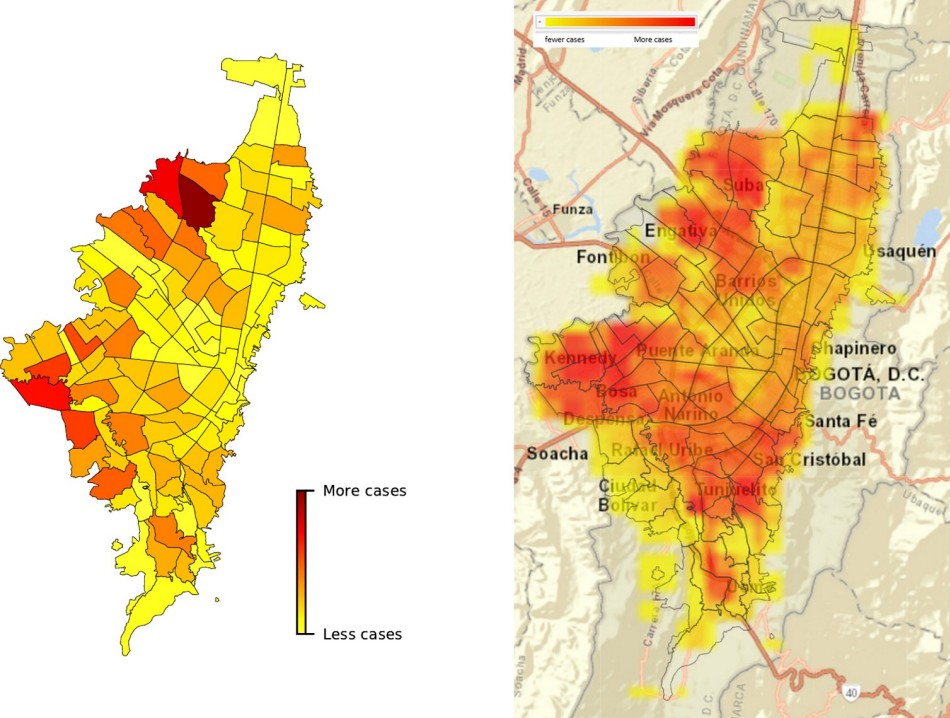

**Fig 7. Comparison between total Seriously-Infected cases in INFEKTA (left) and the real COVID-19 cases confirmed in Bogotá (right).** The colored boxes in the right map (confirmed cases map) corresponds to the concentrations of the cases in 1000 meters on December 30, 2020, in Bogotá.

transmission dynamics of the COVID19, we are able to obtain similar behaviors, in our preliminary experiments, to those reported for the COVID19 in the real world. Therefore, INFEKTA may be able to provide more accurate results if its parameters are set to real ones: disease transmission rates, virus, incubation periods, comorbidity, houses, interest places, routines, population size (close to nine millions of virtual individuals for the Bogotá city).

Despite the usefulness of the INFEKTA, there are some limitations. Since there are fewer people in the model (respect to real people in Bogotá), *Transmilenio* stations would be less crowded than expected and the model could underestimate the transmission of the disease. Further, the model does not cover individual walking from homes to PTS because contact points correspond to a complex network presented in Fig 2. Finally, the routines that each individual has in the simulation are equal day-to-day (i.e., Monday to Sunday with the same routine), this could influence the stochasticity nature of the model.

Our future work will concentrate on studying the transmission of COVID-19 in Bogotá city by considering different scenarios of social distancing rules and by using more realistic information about: i) Relation between personal information and propagation rates of the COVID-19, ii) Places and routes, iii) Population size, and iv) Age and Medical preconditions.

## Supporting information

**S1 File. INFEKTA repository.** A repository containing the source code of the simulator and a technical report explaining the modeling methodology is available at INFEKTA github. (ZIP)

## Author Contributions

**Conceptualization:** Jonatan Gomez, Jeisson Prieto.

**Data curation:** Elizabeth Leon, Arles Rodríguez.

**Investigation:** Jonatan Gomez, Elizabeth Leon, Arles Rodríguez.

**Methodology:** Jonatan Gomez, Jeisson Prieto, Elizabeth Leon, Arles Rodríguez.

**Software:** Jonatan Gomez, Jeisson Prieto.

**Validation:** Jonatan Gomez.

**Visualization:** Jeisson Prieto.

**Writing – original draft:** Jonatan Gomez, Jeisson Prieto, Elizabeth Leon, Arles Rodríguez.

**Writing – review & editing:** Jonatan Gomez, Jeisson Prieto, Elizabeth Leon, Arles Rodríguez.

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
