## [Decision Letter · Decision Letter 0]

28 May 2020

PONE-D-20-11631

INFEKTA: A General Agent-based Model for Transmission of Infectious Diseases: Studying the COVID-19 Propagation in Bogotá - Colombia

PLOS ONE

Dear Dr. Gomez,

Thank you for submitting your manuscript to PLOS ONE. After careful consideration, we feel that it has merit but does not fully meet PLOS ONE’s publication criteria as it currently stands. Therefore, we invite you to submit a revised version of the manuscript that addresses the points raised during the review process.

We look forward to receiving your revised manuscript.

Kind regards,

Rebecca Lee Smith, D.V.M., M.S., Ph.D.

Academic Editor

PLOS ONE

Journal Requirements:

2. Our internal editors have looked over your manuscript and determined that it is within the scope of our Cities as Complex Systems Call for Papers. This collection of papers is headed by a team of Guest Editors for PLOS ONE: Marta Gonzalez (University of California, Berkeley) and Diego Rybski (Potsdam Institute for Climate Impact Research). The Collection will encompass a diverse and interdisciplinary set of research articles applying the principles of complex systems and networks to problems in urban science.  Additional information can be found on our announcement page: https://collections.plos.org/s/cities.

If you would like your manuscript to be considered for this collection, please let us know in your cover letter and we will ensure that your paper is treated as if you were responding to this call. If you would prefer to remove your manuscript from collection consideration, please specify this in the cover letter.

3. We note that Figures in your submission contain [map/satellite] images which may be copyrighted. All PLOS content is published under the Creative Commons Attribution License (CC BY 4.0), which means that the manuscript, images, and Supporting Information files will be freely available online, and any third party is permitted to access, download, copy, distribute, and use these materials in any way, even commercially, with proper attribution. For these reasons, we cannot publish previously copyrighted maps or satellite images created using proprietary data, such as Google software (Google Maps, Street View, and Earth). For more information, see our copyright guidelines: http://journals.plos.org/plosone/s/licenses-and-copyright.

1.    You may seek permission from the original copyright holder of Figures to publish the content specifically under the CC BY 4.0 license.

Additional Editor Comments (if provided):

Reviewers' comments:

Reviewer's Responses to Questions

**Comments to the Author**

1. Is the manuscript technically sound, and do the data support the conclusions?

Reviewer #1: Yes

Reviewer #2: Partly

2. Has the statistical analysis been performed appropriately and rigorously? 

Reviewer #1: N/A

Reviewer #2: I Don't Know

3. Have the authors made all data underlying the findings in their manuscript fully available?

Reviewer #1: Yes

Reviewer #2: Yes

4. Is the manuscript presented in an intelligible fashion and written in standard English?

Reviewer #1: Yes

Reviewer #2: Yes

5. Review Comments to the Author

Reviewer #1: This manuscript describes a novel agent-based model of SARS-CoV-2 transmission. The model includes public transportation, which is a feature frequently ignored in mathematical and computational models. This feature is particularly relevant for the local context of Bogotá, given the potential transmission of SARS-CoV-2 in the local public transportation system (Transmilenio). Transmission in the model occurs through interactions among individuals in different places. The places visited by each individual is based on a trajectory defined for each agent. Once infected, individuals follow a modification of the SEIR model, including seriously-infected and critically-infected individuals. The authors simulated 1001 individuals with demographic characteristics representative of Bogotá. The preliminary analyses from the authors showed that around 20% of the population would be infected within 30 days, and found that social distancing policies reduce the number of infected individuals.

The model presented is interesting and the authors make an effort to include realistic interactions between individuals. However, I have some concerns about the model and the results presented. I hope my comments are useful to the authors to improve their manuscript.

1. Size of the simulations and model capacity

The simulations presented in this study only include 1001 individuals. This is only a small fraction of the total population of Bogotá (8+ million). I'm concerned that this small size does not represent the density of the city. The synthetic population is geographically located according to the population density of the city, but with only 1001 individuals, the density of contacts is much lower than in reality. In this sense, the authors might be simulating a much less dense city than Bogotá. In addition, the small sample size would increase the impact of stochastic effects in the model, so that the epidemic could die out much sooner than expected.. This can explain why the authors found a similar effect of reducing social interactions by 75% than by 95%. I recommend looking into other agent-based models that don't simulate the entire population (https://www.mobs-lab.org/uploads/6/7/8/7/6787877/tracing_main_may4.pdf). Also, multiple simulations of the model could account for these stochastic effects, but the authors do not mention how many replicates were simulated with the same set of parameters.

2. Epidemiological model

The transmission model might not be the appropriate for SARS-CoV-2. As I understand the model, exposed individuals are able to transmit the virus, which doesn't account for the incubation period of the virus. In addition to the incubation period, two important aspects of SARS-CoV-2 transmission should be included: asymptomatic transmission and pre-symptomatic transmission.

3. Place-specific contact rate/transmission probability

It's unclear to me if the transmission rates are the same for each place. If so, this might be unrealistic. The authors should account for place-specific contact rates to determine the transmission probability. For instance, if an individual works in a workplace with 1,000 employees, would INFEKTA consider that these contacts are as relevant for transmission as household contacts are?

4. Simulation experiments

The simulation experiments in this manuscript might not be enough to show the model capabilities. The current simulation experiments only take into consideration social distancing but the authors omit any specific public transportation intervention. Given that including public transportation routes is a strength of this work, the authors could perform some simulations that show the impact of changes in public transportation schedules and capacity. Current policies in Bogotá limit the capacity of transmilenio to about 35% of total capacity. The authors could explore similar strategies. Finally, it isn't clear how transmission occurs in the public transportation. Does it only happen in the terminals or bus routes?

Other comments:

- Abstract: "a Euclidean" should be "an Euclidean"

- Abstract: What are the main results of this study?

- Line 17-18. Try to avoid using "predicting", use estimating or projecting instead

- Lines 24-25. Should "social separation" be "social distancing"?

- Lines 26 - 77. This section might be unnecessarily long, and could be a part of the introduction. More importantly, I suggest that the authors elaborate more on the recent agent-based models available in the literature for COVID-19. It would be helpful to have more context on the current modeling approaches (Examples: Ferguson et al., or Vespigniani et al.) and how INFEKTA compares to those models.

- Lines 99-102. "No home is neighbor to any other home". Does this mean that neighborhood interactions among individuals only occur at public transportation stations?

- Line 145. Please explain Figure 1 with more detail.

- Methods: It isn't clear to me if the parameters for transmission are fixed parameters or drawn from a distribution.

- Methods: It isn't clear to me which processes are random and which are deterministic in the model.

- Methods: Please explain in more detail how the synthetic population was created to match demographic characteristics of Bogotá.

- Lines 188-232: How do the authors account for the difference in the 1001 people in the simulation and the actual population of Bogotá? If the entire city is being simulated but only 1001 people are modeled, how does this affect the transmission dynamics? Is there any factor to account for the difference in synthetic and real density?

- Lines: 286-290. This could be caused by the small sample size. Stochastic effects can make a big difference with only 1001 people simulated.

- Discussion: How does this model compare to other ABMs around the world?

Reviewer #2: 1. In abstract, the sentence “In this paper, a general agent-based model, called…”, I am not sure what authors mean by a general agent-based model. The word general sounds like a specific type of ABM. I suggest to explain what the word “general” means in this context, and if the ABM is a regular ABM, this word in unnecessary.

2. I suggest that authors add at least one more line in the abstract where one central or maybe the most important result of the paper is presented. I believe that usually this attracts reader attention.

3. In introduction, the sentence (line 10) “Complex systems are computational approaches that make use of computer-based algorithms to model dynamic interactions between individuals agents…” is not accurate. Please, revise the complex systems concept. Actually, this sentence make an inversion between complex systems and computer-based algorithms. Complex systems may be simulated by using computer algorithms, but the concept of such systems are wider than the one expressed in this sentence.

4. I think that the subsection “Complex Systems Approaches for Epidemic Models” (line 26) would make more sense if it is locate after the subsection “Compartmental Model” (line 37). Thus, the new sequence would be “Compartmental Model” and after this subsection, “Complex Systems Approaches for Epidemic Models”. My argument is that initially authors should explain what compartmental models are and the limitations they might have. After that, an introduction about complex systems approach an then, the ABM subsection.

5. First paragraph of the subsection “Agent-based models for infectious disease” is fine, although I think authors could add a comment about that agents in an ABM model may take decisions based on some system conditions or rules.

6. Lines 67 to 70 are repeating what was already mentioned about ABMs and complex system, then I suggest, as these lines a redundant, they could be deleted.

7. The sentence in lines 73 to 75 are not very clear to me. Is this what authors meant: In this way, each virtual individual includes its health state, as implemented in a SEIR model, which is fundamental for the infectious disease transmission?

8. In Section “INFEKTA Agent-Based Model”, for the sake of simplicity, the first paragraph should be reduced to “Our agent-based model of infectious disease propagation, called INFEKTA, consists of five-layer components:”. The remaining part of line 80 until line 85 may be delete because each one of the model layers are explained right after the colon. Therefore, I think that the text would benefit of clarity if each layer does not have subsection, maybe items or subitens. For instance, the layer “Space” should explain “Place (Node)” and “Neighbor (Edge)” as part of the same text the item “Space”. The same idea may be applied to explanation of the other remaining layers.

9. Lines 87 to 90: important concepts of the model are presented in these lines. Part of them are between parentheses, so I suggest that they be place between commas, because they are fundamental to understand the concept that is being exposed.

10. Neighbor (Edge), line 98: concept is simple but the relations (neighborhood) among them are not very clear in the explanation. Authors may consider two options: re-write the sentence or add a simple figure representing each concept and the relations among them.

11. Line 100, the footnote: replace "in" by "In".

12. Line 119. What is “Going hour”? There is no definition for it in the text.

13. Line 135. In the case of covid-19, as literally everybody is susceptible to the SARS-Cov-2 virus, the health state “Immune (M)” only makes sense when the individual is recovered from the disease. However, the “Recovered (R)” health state is also defined in the model. Then, my question is: what is the difference between the two health states? Besides, in line 142, authors say “INFEKTA introduces both the M state since some individuals are naturally immune”. In the literature, one can find that, if so, only a few persons have some partial immunity against SARS-Cov (not SARS-Cov-2), however, this is not a lifetime imunnity and this outbreak happened in the early 2000’s. Moreover, the amount of people infected with SARS-Cov is so small and geographically restricted that does not justify to model a pandemic caused by SARS-Cov-2 considering that some individuals have any prior immunity. Then, this part of the text has to be re-written.

Figure 1 shows clearly the immunity is acquired only after the individual has recovered from the infection: this is the correct dynamics of covid-19 outbreak and, as recovered individuals have acquired immunity immune, a new compartment called “M” is not necessary.

In addition, in Fig.1, there is a line from R compartment to S compartment, meaning that recovered individuals become susceptible again. So far, in the literature about covid-19 pandemic there is no evidence that recovered individuals become susceptible again. Thus, this fact implies that this model has to be considered incorrect and for this reason, until this correction is made, I suggest the paper is rejected.

6. PLOS authors have the option to publish the peer review history of their article (what does this mean?). If published, this will include your full peer review and any attached files.

Reviewer #1: Yes: Guido Camargo España

Reviewer #2: No

---

## [Author Response · Author response to Decision Letter 0]

27 Oct 2020

Reviewer #1: 

We thank the reviewer for appreciating our paper updates and for providing further constructive suggestions on how to improve the next version. We hope to have addressed these in a satisfactory manner, as explained below, point-by-point.

1. Size of the simulations and model capacity: The simulations presented in this study only include 1001 individuals. This is only a small fraction of the total population of Bogotá (8+ million). I'm concerned that this small size does not represent the density of the city. The synthetic population is geographically located according to the population density of the city, but with only 1001 individuals, the density of contacts is much lower than in reality. In this sense, the authors might be simulating a much less dense city than Bogotá. In addition, the small sample size would increase the impact of stochastic effects in the model, so that the epidemic could die out much sooner than expected.. This can explain why the authors found a similar effect of reducing social interactions by 75% than by 95%. I recommend looking into other agent-based models that don't simulate the entire population (https://www.mobs-lab.org/uploads/6/7/8/7/6787877/tracing_main_may4.pdf). Also, multiple simulations of the model could account for these stochastic effects, but the authors do not mention how many replicates were simulated with the same set of parameters.

+ We followed this recommendation and performed new experiments with 998.213 agents. Bogotá has a population of 7.412.566 and it is composed by 112 Zonal Planning Units (UPZ, for its acronym in spanish). Each UPZ belongs to one of the 19 urban districts in Bogotá. The number of places in each UPZ was generated based on the population density of each UPZ according to the data available in 2017 obtained from public information available by the District Planning Secretary: http://www.sdp.gov.co/gestion-estudios-estrategicos/estudios-macro/proyecciones-de-poblacion.

The following idea was added to clarify this query: The number of places in each UPZ is generated based on the population density of each UPZ according to the data available in 2017 [27]. Table 1 shows the amount of data generated for each type of place and for people, also the number of TM stations (Bus), and terminal transportation that we use in the simulation, and Table 2 shows a detailed information of the number of interest places generated by UPZ.

2. Epidemiological model

The transmission model might not be the appropriate for SARS-CoV-2. As I understand the model, exposed individuals are able to transmit the virus, which doesn't account for the incubation period of the virus. In addition to the incubation period, two important aspects of SARS-CoV-2 transmission should be included: asymptomatic transmission and pre-symptomatic transmission.

+ We have included a more detailed description of the model in section Agents based model for infectious disease and include a clock symbol that indicates a period of time able to change to another state. The reviewer is right. We include the Asymptomatic-Infected (IA) state after the exposed one to model the incubation period of the virus. 

3. Place-specific contact rate/transmission probability

It's unclear to me if the transmission rates are the same for each place. If so, this might be unrealistic. The authors should account for place-specific contact rates to determine the transmission probability. For instance, if an individual works in a workplace with 1,000 employees, would INFEKTA consider that these contacts are as relevant for transmission as household contacts are?

+ We model fixed probabilities to model changes of state of an individual. Probabilities are applied if two individuals are close enough in the same place or transportation station. It means that more dense places implies more possible infected individuals.

In section Agents based model for infectious disease we added: 

Each individual is modelled as an agent with an internal ”SEIR” state that represents its infectious disease state (severity and time in it) at any instant of time. Individuals interact between them, i.e., can infect or get infected, when they move at some instant to the same place, their SEIR states, and the infectious disease transmission rates, and how close they are (if they are in crowded places). Notice, the concept of crowded places is natural in INFEKTA and emerges from the agents interactions (eg. if more individuals move using the same transportation’s routes) and individual characteristics (e.g. children go to schools.).

4. Simulation experiments

The simulation experiments in this manuscript might not be enough to show the model capabilities. The current simulation experiments only take into consideration social distancing but the authors omit any specific public transportation intervention. Given that including public transportation routes is a strength of this work, the authors could perform some simulations that show the impact of changes in public transportation schedules and capacity. Current policies in Bogotá limit the capacity of transmilenio to about 35% of total capacity. The authors could explore similar strategies. Finally, it isn't clear how transmission occurs in the public transportation. Does it only happen in the terminals or bus routes?

+ Indeed, this aspect wasn’t clear in the paper and we thank the reviewer for pointing it out. By now, we don't cover the capacity limit of transmilenio but model individual dynamics. As in the previous answer we add the following idea: Notice, the concept of crowded places is natural in INFEKTA and emerges from the agents interactions (eg. if more individuals move using the same transportation’s routes) and individual characteristics (e.g. children go to schools.).

In the section individual setup we added the following: 

Also, a sequence of activities was assigned randomly to each individual to define a diary routine. This was done according to the person’s age and the hour of the day. For example, some agents Adolescence go to school, and some agents Adult go to work. Time to start routine -going from Home to IP and return- is randomly selected in the interval from 4h and 7h returning between the 17h and 20h.

Other comments:

- Abstract: "a Euclidean" should be "an Euclidean" (fixed, thank you)

- Abstract: What are the main results of this study? 

+ The abstract was rewritten and main results of the study were included: This paper proposes an agent-based model, called INFEKTA, for simulating the transmission of infectious diseases, not only the COVID19, under social distancing policies.INFEKTA combines the transmission dynamic of an specific disease, (according to parameters found in the literature) with demographic information (population density, age, and genre of individuals) of geopolitical regions of the real town or city under study. Agents (virtual persons) can move, according to its mobility routines and the enforced social distancing policy, on a complex network of accessible places defined over an Euclidean space representing the town or city. The transmission dynamics of the COVID-19 under different social distancing policies in Bogotá city, the capital of Colombia, is simulated using INFEKTA with one million virtual persons. A sensitivity analysis of the impact of those social distancing policies on the disease rates (Asymptomatic, Seriously, and Critically) indicates that it is possible to establish a not so ’hard’ social distancing policy to achieve a significant reduction on the disease rate.

- Line 17-18. Try to avoid using "predicting", use estimating or predicting instead (fixed, thank you)

- Lines 24-25. Should "social separation" be "social distancing"? (fixed, thank you)

- Lines 26 - 77. This section might be unnecessarily long, and could be a part of the introduction. More importantly, I suggest that the authors elaborate more on the recent agent-based models available in the literature for COVID-19. It would be helpful to have more context on the current modeling approaches (Examples: Ferguson et al., or Vespigniani et al.) and how INFEKTA compares to those models.

+ Some other agent models were added in the following paragraph: In the past, ABMs have been employed to address various infectious diseases such as, a bioterrorist introduction of smallpox [9], control of tuberculosis [10], implementation of distancing measures and antiviral prophylaxis to control H5N1 influenza A (bird20flu) [11], design of vaccination strategies for influenza [12], devise evacuation strategies in the event of airborne contamination [13], and curtail transmission of measles through contact tracing and quarantine [14]. While this paper was under revision, some other works that include heterogeneous agents and social distancing were proposed to model COVID-19 [15, 16]. The proposed approach, called INFEKTA (Esperanto word for infectious), mainly differs from existing works in that it aims to generate individuals and a complex network of places based on the population density of a determined city.

- Lines 99-102. "No home is neighbor to any other home". Does this mean that neighborhood interactions among individuals only occur at public transportation stations? 

+ The idea was extended as follows: Each individual has a Home and an IP. The closest distances are computed between each Home and IP and between these places with the closest PTS using their longitude and latitude. If the distance between a Home and IP is shorter than the distance to a PTS this Home will be connected directly to an IP instead of their closest PTS. Otherwise, the closest PTS is connected to each Home and IP respectively. Detailed information can be found in the Figure 2 of Section Virtual Space Setup.

- Line 145. Please explain Figure 1 with more detail.

+ Description of Figure 1, was extended and also the section to give more details to the model.

- Methods: It isn't clear to me if the parameters for transmission are fixed parameters or drawn from a distribution. 

Parameters are fixed values extracted from a literature review. 

+ We wrapped them in the transition rates and allowed modelers to change and play with different rates. Therefore, we set the initial values of these rates as shown in Table 4.

- Methods: It isn't clear to me which processes are random and which are deterministic in the model.

To address this comment, we added Table 4 that describes the initial parameters of Infekta based on a literature review. + Probabilities and distribution of individuals are random and specific times are deterministic.

- Methods: Please explain in more detail how the synthetic population was created to match demographic characteristics of Bogotá. 

+ We have clarified this aspect in the reply to comment #1. 

- Lines 188-232: How do the authors account for the difference in the 1001 people in the simulation and the actual population of Bogotá? If the entire city is being simulated but only 1001 people are modeled, how does this affect the transmission dynamics? Is there any factor to account for the difference in synthetic and real density?

We have clarified this aspect in the reply to comment #1. 

- Lines: 286-290. This could be caused by the small sample size. Stochastic effects can make a big difference with only 1001 people simulated.

+ We have clarified this aspect in the reply to comment #1. We thank the reviewer for this. 

- Discussion: How does this model compare to other ABMs around the world?

+ Some other agent models were added in the following paragraph: In the past, ABMs have been employed to address various infectious diseases such as, a bioterrorist introduction of smallpox [9], control of tuberculosis [10], implementation of distancing measures and antiviral prophylaxis to control H5N1 influenza A (bird20flu) [11], design of vaccination strategies for influenza [12], devise evacuation strategies in the event of airborne contamination [13], and curtail transmission of measles through contact tracing and quarantine [14]. While this paper was under revision, some other works that include heterogeneous agents and social distancing were proposed to model COVID-19 [15, 16]. The proposed approach, called INFEKTA (Esperanto word for infectious), mainly differs from existing works in that it aims to generate individuals and a complex network of places based on the population density of a determined city.

Reviewer #2: 

We thank the reviewer for appreciating our paper updates and for providing further constructive suggestions on how to improve the next version. We hope to have addressed these in a satisfactory manner, as explained below, point-by-point.

1. In abstract, the sentence “In this paper, a general agent-based model, called…”, I am not sure what authors mean by a general agent-based model. The word general sounds like a specific type of ABM. I suggest to explain what the word “general” means in this context, and if the ABM is a regular ABM, this word is unnecessary.

+ Word general has been removed from the abstract. And clarify this point 

This paper proposes an agent-based model, called INFEKTA, for simulating the transmission of infectious diseases, not only the COVID19, under social distancing policies. INFEKTA combines the transmission dynamic of an specific disease, (according to parameters found in the literature) with demographic information (population density, age, and genre of individuals) of geopolitical regions of the real town or city under study. Agents (virtual persons) can move, according to its mobility routines and the enforced social distancing policy, on a complex network of accessible places defined over an Euclidean space representing the town or city. 

2. I suggest that authors add at least one more line in the abstract where one central or maybe the most important result of the paper is presented. I believe that usually this attracts reader attention.

+ Thanks for this suggestion we rewrite the abstract as follows: This paper proposes an agent-based model, called INFEKTA, for simulating the transmission of infectious diseases, not only the COVID19, under social distancing policies.INFEKTA combines the transmission dynamic of an specific disease, (according to parameters found in the literature) with demographic information (population density, age, and genre of individuals) of geopolitical regions of the real town or city under study. Agents (virtual persons) can move, according to its mobility routines and the enforced social distancing policy, on a complex network of accessible places defined over an Euclidean space representing the town or city. The transmission dynamics of the COVID-19 under different social distancing policies in Bogotá city, the capital of Colombia, is simulated using INFEKTA with one million virtual persons. A sensitivity analysis of the impact of those social distancing policies on the disease rates (Asymptomatic, Seriously, and Critically) indicates that it is possible to establish a not so ’hard’ social distancing policy to achieve a significant reduction on the disease rate.

3. In introduction, the sentence (line 10) “Complex systems are computational approaches that make use of computer-based algorithms to model dynamic interactions between individuals agents…” is not accurate. Please, revise the complex systems concept. Actually, this sentence makes an inversion between complex systems and computer-based algorithms. Complex systems may be simulated by using computer algorithms, but the concept of such systems are wider than the one expressed in this sentence.

+ Indeed, the reviewer is right. We state the following: Many biological systems have been modeled in terms of complex systems since their collective behavior cannot be simply inferred from the understanding of their components.

Additionally, We replaced “Complex systems are computational approaches” for “Complex systems may be simulated by using computer-based algorithms to model dynamic interactions between individuals agents (e.g. persons, cells) or groups and their properties, within, and across levels of influence [6, 7].” 

4. I think that the subsection “Complex Systems Approaches for Epidemic Models” (line 26) would make more sense if it is locate after the subsection “Compartmental Model” (line 37). Thus, the new sequence would be “Compartmental Model” and after this subsection, “Complex Systems Approaches for Epidemic Models”. My argument is that initially authors should explain what compartmental models are and the limitations they might have. After that, an introduction about complex systems approach an then, the ABM subsection.

+ Since for our model the most important part is the complex network of places more than the compartmental model, we decide to adopt the current structure. We added examples and Figures like Figure 3 to clarify why we adopt this subsection structure.

5. First paragraph of the subsection “Agent-based models for infectious disease” is fine, although I think authors could add a comment about that agents in an ABM model may take decisions based on some system conditions or rules.

+ In section Agents based model for infectious disease we added: 

Each individual is modelled as an agent with an internal ”SEIR” state that represents its infectious disease state (severity and time in it) at any instant of time. Individuals interact between them, i.e., can infect or get infected, when they move at some instant to the same place, their SEIR states, and the infectious disease transmission rates, and how close they are (if they are in crowded places). Notice, the concept of crowded places is natural in INFEKTA and emerges from the agents interactions (eg. if more individuals move using the same transportation’s routes) and individual characteristics (e.g. children go to schools.).

6. Lines 67 to 70 are repeating what was already mentioned about ABMs and complex system, then I suggest, as these lines a redundant, they could be deleted. (fixed, thank you)

7. The sentence in lines 73 to 75 are not very clear to me. Is this what authors meant: In this way, each virtual individual includes its health state, as implemented in a SEIR model, which is fundamental for the infectious disease transmission?

+ We added the paragraph presented as the answer to comment 5 to clarify the idea.

8. In Section “INFEKTA Agent-Based Model”, for the sake of simplicity, the first paragraph should be reduced to “Our agent-based model of infectious disease propagation, called INFEKTA, consists of five-layer components:”. The remaining part of line 80 until line 85 may be delete because each one of the model layers are explained right after the colon. Therefore, I think that the text would benefit of clarity if each layer does not have subsection, maybe items or subitens. For instance, the layer “Space” should explain “Place (Node)” and “Neighbor (Edge)” as part of the same text the item “Space”. The same idea may be applied to explanation of the other remaining layers. (fixed, thank you)

9. Lines 87 to 90: important concepts of the model are presented in these lines. Part of them are between parentheses, so I suggest that they be place between commas, because they are fundamental to understand the concept that is being exposed. (fixed, thank you)

10. Neighbor (Edge), line 98: concept is simple but the relations (neighborhood) among them are not very clear in the explanation. Authors may consider two options: re-write the sentence or add a simple figure representing each concept and the relations among them.

+ The sentence was clarified as follows: Each individual has a Home and an IP. The closest distances are computed between each Home and IP and between these places with the closest PTS using their longitude and latitude. If the distance between a Home and IP is shorter than the distance to a PTS this Home will be connected directly to an IP instead of their closest PTS. Otherwise, the closest PTS is connected to each Home and IP respectively. Detailed information can be found in the Figure 2 of Section Virtual Space Setup.

11. Line 100, the footnote: replace "in" by "In". (fixed, thank you)

12. Line 119. What is “Going hour”? There is no definition for it in the text.

Definition was added: Going hour is the time an individual moves from Home to a IP. In section individuals setup the following was added: Time to start routine -going from Home to IP and return- is randomly selected in the interval from 4h and 7h returning between the 17h and 20h. 

13. Line 135. In the case of covid-19, as literally everybody is susceptible to the SARS-Cov-2 virus, the health state “Immune (M)” only makes sense when the individual is recovered from the disease. However, the “Recovered (R)” health state is also defined in the model. Then, my question is: what is the difference between the two health states? Besides, in line 142, authors say “INFEKTA introduces both the M state since some individuals are naturally immune”. In the literature, one can find that, if so, only a few persons have some partial immunity against SARS-Cov (not SARS-Cov-2), however, this is not a lifetime imunnity and this outbreak happened in the early 2000’s. Moreover, the amount of people infected with SARS-Cov is so small and geographically restricted that does not justify to model a pandemic caused by SARS-Cov-2 considering that some individuals have any prior immunity. Then, this part of the text has to be re-written.

 + Figure 1 shows clearly the immunity is acquired only after the individual has recovered from the infection: this is the correct dynamics of covid-19 outbreak and, as recovered individuals have acquired immunity immune, a new compartment called “M” is not necessary.

In addition, in Fig.1, there is a line from R compartment to S compartment, meaning that recovered individuals become susceptible again. So far, in the literature about covid-19 pandemic there is no evidence that recovered individuals become susceptible again. Thus, this fact implies that this model has to be considered incorrect and for this reason, until this correction is made, I suggest the paper is rejected.

In the abstract we added the following: This paper proposes an agent-based model, called INFEKTA, for simulating the transmission of infectious diseases, not only the COVID19, under social distancing policies. When presenting Figure 1 we clarified this aspect mentioned by the reviewer: Figure 1 shows the general transition dynamics of any infectious disease at the individual level in INFEKTA. This model can be adjusted to any specific infectious disease by setting some of the probabilities to specific values. For example, if there is no evidence that recovered individuals become immune or susceptible again, such probabilities can be set to 0.0.

---

## [Decision Letter · Decision Letter 1]

9 Dec 2020

PONE-D-20-11631R1

INFEKTA: An Agent-based Model for Transmission of Infectious Diseases: Studying the COVID-19 Propagation in Bogotá - Colombia

PLOS ONE

Dear Dr. Gomez,

Thank you for submitting your manuscript to PLOS ONE. After careful consideration, we feel that it has merit but does not fully meet PLOS ONE’s publication criteria as it currently stands. Therefore, we invite you to submit a revised version of the manuscript that addresses the points raised during the review process.

ACADEMIC EDITOR: Please address the small changes suggested by Reviewer 1.

We look forward to receiving your revised manuscript.

Kind regards,

Rebecca Lee Smith

Academic Editor

PLOS ONE

Reviewers' comments:

Reviewer's Responses to Questions

**Comments to the Author**

1. If the authors have adequately addressed your comments raised in a previous round of review and you feel that this manuscript is now acceptable for publication, you may indicate that here to bypass the “Comments to the Author” section, enter your conflict of interest statement in the “Confidential to Editor” section, and submit your "Accept" recommendation.

Reviewer #1: (No Response)

Reviewer #2: All comments have been addressed

2. Is the manuscript technically sound, and do the data support the conclusions?

Reviewer #1: Yes

Reviewer #2: Yes

3. Has the statistical analysis been performed appropriately and rigorously? 

Reviewer #1: N/A

Reviewer #2: Yes

4. Have the authors made all data underlying the findings in their manuscript fully available?

Reviewer #1: Yes

Reviewer #2: Yes

5. Is the manuscript presented in an intelligible fashion and written in standard English?

Reviewer #1: Yes

Reviewer #2: Yes

6. Review Comments to the Author

Reviewer #1: The authors addressed most of my comments in an acceptable manner. Below some additional comments.

First, I apologize for my misguided comment on `a Eucledian`. It is indeed spelled `a Eucledian`, not `an Eucledian`.

- There is still an issue with my first comment regarding the stochasticity nature of the model and the population size. In terms of the stochasticity, the authors only report the mean of their results, do not include any uncertainty. For the population size, I understand the need for simplification in the population size, but the authors should at least add in the discussion what impact does this underestimation have in their results. For instance, transmilenio buses or stations would be less crowded than expected, given that there are fewer people in the model. In general, the authors should add a section in the discussion listing the model limitations.

- The authors added a couple of other models in their introduction, which I appreciate. I don't agree that this model differentiates in having a complex network of places and population density. Several of the agent-based models that have been available for a long time to simulate COVID-19 include realistic representations of human activities and their population demographic characteristics. Including public transportation might be a more unique characteristic of this model. Also, there are many more than two abm for COVID-19, and have been released before this manuscript was under review.

- For future reviews. Please add the line numbers where the changes have been made. It is difficult to follow the responses without the lines where the changes were made.

- Abstract: `not so hard` -> please use specific and descriptive words.

- Line 306-309. the statement "indicate that INFEKTA may be a valuable asset for researchers and public health decision-makers for projecting fugure scenarios when applying social distancing policy rules" is not supported by their results. 1. The authors do not contrast their simulations to any data in the manuscript to calibrate model parameters, 2. The authors do not make any projections and contrast them with epidemiological data in Bogotá. If the model is not validated with data, how can it be used to project future scenarios? I think the model could be used to explore scenarios, but not to project or predict.

Reviewer #2: As the authors have addressed all the suggestions and requested corrections, I think the paper is suitable to be published.

7. PLOS authors have the option to publish the peer review history of their article (what does this mean?). If published, this will include your full peer review and any attached files.

Reviewer #1: No

Reviewer #2: **Yes: **Aquino L. Espindola

---

## [Author Response · Author response to Decision Letter 1]

6 Jan 2021

Dear Editor,

First, we would like to thank you for providing further constructive suggestions on how to improve the next version. We provide a few answers and clarifications for each outstanding comment below. Answers to the queries are in blue and literal changes added to the paper are in red.

Reviewer #1: 

Many thanks for providing further constructive suggestions on how to improve the next version. We hope to have addressed these in a satisfactory manner, as explained below, point-by-point.

1. There is still an issue with my first comment regarding the stochasticity nature of the model and the population size. In terms of the stochasticity, the authors only report the mean of their results, do not include any uncertainty. For the population size, I understand the need for simplification in the population size, but the authors should at least add in the discussion what impact does this underestimation have in their results. For instance, transmilenio buses or stations would be less crowded than expected, given that there are fewer people in the model. In general, the authors should add a section in the discussion listing the model limitations.

- Standard deviation data were added to Figures 4, 5 and 6 that corresponds to the experiments performed.

- Also, we add a paragraph with the limitations in lines 250-256: Despite the usefulness of the INFEKTA, there are some limitations. Since there are fewer people in the model (respect to real people in Bogotá), Transmilenio stations would be less crowded than expected and the model could underestimate the transmission of the disease. Further, the model does not cover individual walking from homes to PTS because contact points correspond to a complex network presented in Figure 2. Finally, the routines that each individual has in the simulation are equal day-to-day (i.e., Monday to Sunday with the same routine), this could influence the stochasticity nature of the model.

2. The authors added a couple of other models in their introduction, which I appreciate. I don't agree that this model differentiates in having a complex network of places and population density. Several of the agent-based models that have been available for a long time to simulate COVID-19 include realistic representations of human activities and their population demographic characteristics. Including public transportation might be a more unique characteristic of this model. Also, there are many more than two abm for COVID-19, and have been released before this manuscript was under review.

- Thank you for this comment. Lines 23 - 28 were updated: In our literature review, some other novel works that include heterogeneous agents and social distancing were proposed to model COVID-19 [15],[16]. The proposed approach, called INFEKTA (Esperanto word for infectious), mainly differs from existing works in that it aims to generate individuals and a complex network of places based on the population density of a determined city including individual interaction in public transportation means.

3. For future reviews. Please add the line numbers where the changes have been made. It is difficult to follow the responses without the lines where the changes were made. 

- (Thanks for your advice)

4. Abstract: `not so hard` -> please use specific and descriptive words.

- At the end of the Abstract we change the word ‘hard’ by ‘medium’ and give the corresponding definition (The full description of the ‘medium’ social-distance policy is in the line 251, section Social Distancing Rule Setting): A sensitivity analysis of the impact of social distancing policies indicates that it is possible to establish a 'medium' (i.e., close 40\\% of the places) social distancing policy to achieve a significant reduction in the disease transmission.

5. Line 306-309. the statement "indicate that INFEKTA may be a valuable asset for researchers and public health decision-makers for projecting future scenarios when applying social distancing policy rules" is not supported by their results. 1. The authors do not contrast their simulations to any data in the manuscript to calibrate model parameters, 2. The authors do not make any projections and contrast them with epidemiological data in Bogotá. If the model is not validated with data, how can it be used to project future scenarios? I think the model could be used to explore scenarios, but not to project or predict.

- Word projecting was changed by exploring (lines 325-328) and figure 7 was added as shown in the paper (lines 297-306):

- Although our intention was not to predict geographic spread for the city, we observed similarities between the total Seriously-Infected cases in INFEKTA (we assume that individuals in Seriously-Infected state are cases tested in Bogotá) and the current concentration of COVID-19 cases confirmed in Bogotá, see Figure 7. The results show how UPZ with more cases found with INFEKTA matches with geographic areas with more COVID-19 cases. Then, how population distribution is generated from real density data of Bogotá, INFEKTA can be useful to explore policies by Zonal Planning Units (UPZ) or territorial divisions of a selected place providing to recommend actions for before, during, and after pandemic i.e., in planning and coordination efforts through leadership and coordination across sectors.

---

## [Editor Report · Decision Letter 2]

8 Jan 2021

INFEKTA: An Agent-based Model for Transmission of Infectious Diseases: The COVID-19 case in Bogotá, Colombia

PONE-D-20-11631R2

Dear Dr. Gomez,

We’re pleased to inform you that your manuscript has been judged scientifically suitable for publication and will be formally accepted for publication once it meets all outstanding technical requirements.

Kind regards,

Rebecca Lee Smith, D.V.M., M.S., Ph.D.

Academic Editor

PLOS ONE
---

## [Editor Report · Acceptance letter]

29 Jan 2021

PONE-D-20-11631R2 

INFEKTA - An Agent-based Model for Transmission of Infectious Diseases: The COVID-19 case in Bogotá, Colombia  

Dear Dr. Gomez:

I'm pleased to inform you that your manuscript has been deemed suitable for publication in PLOS ONE. Congratulations! Your manuscript is now with our production department. 

Kind regards, 

on behalf of

Dr. Rebecca Lee Smith 

Academic Editor

PLOS ONE